# A Facile Strategy for the Preparation of N-Doped TiO_2_ with Oxygen Vacancy via the Annealing Treatment with Urea

**DOI:** 10.3390/nano14100818

**Published:** 2024-05-07

**Authors:** Zhe Zhang, Zhenpeng Cui, Yinghao Xu, Mohamed Nawfal Ghazzal, Christophe Colbeau-Justin, Duoqiang Pan, Wangsuo Wu

**Affiliations:** 1School of Nuclear Science and Technology, Lanzhou University, Lanzhou 730000, China; 2Frontiers Science Center for Rare Isotopes, Lanzhou University, Lanzhou 730000, China; 3Institute of Physical Chemistry, Paris-Saclay University, 91405 Orsay, France

**Keywords:** nitrogen doping, TiO_2_, oxygen vacancy, annealing treatment, urea

## Abstract

Although titanium dioxide (TiO_2_) has a wide range of potential applications, the photocatalytic performance of TiO_2_ is limited by both its limited photoresponse range and fast recombination of the photogenerated charge carriers. In this work, the preparation of nitrogen (N)-doped TiO_2_ accompanied by the introduction of oxygen vacancy (Vo) has been achieved via a facile annealing treatment with urea as the N source. During the annealing treatment, the presence of urea not only realizes the N-doping of TiO_2_ but also creates Vo in N-doped TiO_2_ (N-TiO_2_), which is also suitable for commercial TiO_2_ (P25). Unexpectedly, the annealing treatment-induced decrease in the specific surface area of N-TiO_2_ is inhibited by the N-doping and, thus, more active sites are maintained. Therefore, both the N-doping and formation of Vo as well as the increased active sites contribute to the excellent photocatalytic performance of N-TiO_2_ under visible light irradiation. Our work offers a facile strategy for the preparation of N-TiO_2_ with Vo via the annealing treatment with urea.

## 1. Introduction

Titanium dioxide (TiO_2_) has been widely investigated because of its excellent chemical stability, nontoxicity, and low cost [1,2]. Although TiO_2_ shows great potential in many fields [3,4,5], it can only absorb ultraviolet (UV) light [6], which limits its sufficient absorption of solar light because visible light (43%) takes up the majority of the solar spectrum [7]. In addition, the fast recombination of photogenerated charge carriers in TiO_2_ impedes its efficient use of solar energy [8,9]. Thus, in order to improve the photocatalytic performance of TiO_2_, not only does the photoresponse range need to be extended but also the recombination of photogenerated charge carriers needs to be inhibited [10,11].

To extend the photoresponse range of TiO_2_ from UV light to visible light, the doping strategy is commonly applied to adjust the intrinsic wide bandgap of TiO_2_ [12,13]. Among various doping strategies, it is reported that nitrogen (N) doping is an effective approach to reduce the bandgap of TiO_2_ and enables its absorption of visible light [14,15,16]. Generally, the preparation of N-doped TiO_2_ can be realized by the annealing treatment with the presence of additional N sources such as urea and ammonia [17,18,19]. To reduce the recombination of photogenerated charge carriers, the formation of defective structure is reported to be a useful strategy [6,20]. The introduction of point defects into TiO_2_, such as oxygen vacancy (Vo), could trap the photogenerated electron inhibiting the recombination of the photogenerated charge carriers [10,21]. Although many methods have been reported for the creation of Vo in TiO_2_ [22,23,24], the annealing treatment of TiO_2_ with an organic additive is reported to be an easy approach for the creation of Vo [25,26]. Thus, it is of great interest to realize both the N-doping and creation of Vo to enhance the photocatalytic performance of TiO_2_.

To realize both the N-doping and introduction of Vo in one step, in this work, a facile strategy has been developed to prepare N-doped TiO_2_ accompanied by the formation of Vo via the annealing treatment with urea. The effects of the annealing treatment on the photoresponse property and crystal structure of TiO_2_ have been investigated as a function of the amount of urea. The photocatalytic performance of as-synthesized TiO_2_ photocatalysts has been evaluated by the photocatalytic degradation of organic pollutants under visible light irradiation.

## 2. Materials and Methods

### 2.1. Materials

Hydrothermally prepared TiO_2_ was obtained according to the reported synthetic procedures [27]. Commercially available TiO_2_ (P25, Degussa, Evonik, Resource Effiency GmbH, Essen, Germany) was purchased and used without further treatment. Custom-built high-borosilicate glass tubes were used for the annealing treatment. Methyl orange (MO, AR grade, Beijing Chemical Plant Co., Beijing City, China) was chosen as the model organic pollutant for evaluating the photocatalytic performance of all photocatalysts.

### 2.2. Annealing Treatment

To prepare the N-doped TiO_2_ via the annealing treatment, both hydrothermally prepared TiO_2_ and urea were sealed in high-borosilicate glass tubes under vacuum and annealed at 500 °C for 2 h. The amount of TiO_2_ (100 mg) was kept the same and the weight ratios of TiO_2_ to urea were set as 1:0, 2:1, 1:1, and 1:2 (Appendix A). The obtained samples were named as A-TiO_2_, N-TiO_2_ (2:1), N-TiO_2_ (1:1), and N-TiO_2_ (1:2), respectively. Similarly, the annealing treatment of P25 with urea was conducted under the same conditions (Appendix A). The obtained samples were abbreviated as A-P25, N-P25 (2:1), N-P25 (1:1), and N-P25 (1:2), respectively (Appendix A). For comparison purposes, the annealing treatment of urea (100 mg) was also performed under the same conditions (Appendix A). All the as-synthesized samples were collected for further characterizations and tests.

### 2.3. Characterizations

The diffuse reflectance spectra (DRS) were measured by the UV-vis diffuse reflectance spectroscopy (UV-2600, Shimadzu, Kyoto, Japan). The diffraction patterns were recorded by X-ray diffraction (XRD, Ultima IV, Rigaku, Tokyo, Japan) with a Cu-Kα radiation source (λ = 1.5406 Å). Brunauer–Emmett–Teller (BET) surface areas were determined with an accelerated surface area and porosity analyzer (ASAP 2460, Micrometrics, Norcross, GA, USA). Electron paramagnetic resonance (EPR, ER200DSRC, Bruker, Mannheim, Germany) spectra were taken by applying an X-band (9.44 GHz, 2.47 mW) microwave and sweeping magnetic field at room temperature. The ultraviolet-visible (UV-vis) spectra were obtained with a UV-vis spectrophotometer (Lambda 35, PerkinElmer, Waltham, MA, USA).

### 2.4. Photocatalytic Performance

Photocatalytic degradation of MO was carried out under visible light irradiation (Xenon lamp, 300 W, PerfectLight, Beijing, China) with a long-wavelength pass filter (>420 nm). TiO_2_ photocatalysts (25 mg, 1 mg/mL) were added to aqueous solutions of MO (25 ppm, 25 mL) and stirred in dark for 60 min. The concentration variation of MO as a function of irradiation time was monitored by measuring its characteristic peak centered at 464 nm. The photocatalytic degradation ratio of MO was estimated by the expression: (C_0_ − C)/C_0_ × 100%, where C_0_ is the initial concentration of MO and C corresponds to the concentration of MO at different time intervals.

## 3. Results and Discussion

### 3.1. Characterizations of TiO_2_ Photocatalysts

As shown in Figure 1, hydrothermally prepared TiO_2_ appears as a white powder before the annealing treatment (Figure 1a and Appendix A). In the absence of urea, TiO_2_ turns to a dark grey powder after the annealing treatment (Figure 1b and Appendix A) due to the pyrolysis of the organic solvent [27,28]. With the presence of urea (Appendix A), TiO_2_ is transformed to a brown powder after the annealing treatment (Appendix A). This obvious color change indicates that N-doping of TiO_2_ may happen during the annealing treatment with urea as the N source [29,30]. Although the amount of urea increases, the colors of all N-TiO_2_ samples are similar, implying that a low amount of urea is enough for N-doping (Figure 1c–e). This facile strategy is also suitable for the preparation of N-doped P25 (Appendix A) and white P25 changes to a brown powder after the annealing treatment with urea (Appendix A). Since the annealing treatment results in no clear color change of urea (Appendix A), it is reasonable to propose that the pyrolysis of urea leads to the N-doping of TiO_2_ during the annealing treatment.

To investigate the effects of the annealing treatment on the photoresponse property of TiO_2_, the DRS spectra of all samples were recorded. According to Figure 2a, both TiO_2_ and A-TiO_2_ show clear UV absorption and a negligible visible light response. This result is in agreement with the color of TiO_2_ and the reported phenomenon [27]. Conversely, all N-TiO_2_ samples present obvious visible light absorption with almost identical absorbance, which further proves that a low amount of urea is enough for N-doping. The annealing treatment of TiO_2_ with urea definitely expands the photoresponse range of TiO_2_ to the visible light range suggesting that N-doping occurs [13,31]. To further confirm the N-doping of TiO_2_, N1s XPS fine spectra of all samples were collected. Compared with TiO_2_ and A-TiO_2_, all N-TiO_2_ samples show a clear N1s peak which undoubtedly proves that the N-doping of TiO_2_ occurs [32,33]. In addition, the peak intensity increases as the amount of urea goes up, implying an increase in the N-doping level [18,34]. Thus, the annealing treatment of urea offers a facile approach for the preparation of N-doped TiO_2_ to extend the photoresponse range of TiO_2_.

To further study the influences of the annealing treatment on the crystal structures of TiO_2_, XRD measurements were conducted and the results are shown in Figure 3. All the characteristic diffraction patterns of TiO_2_ photocatalysts are in accordance with the diffraction peaks of anatase, proving that the crystal phase of all TiO_2_ samples is anatase (JCPDS-21-1272) [25,35]. In the absence of urea, the crystallinity of TiO_2_ is improved after the annealing treatment as indicated by the increased intensity of the diffraction patterns which correspond to the (101) and (200) crystal planes. However, the crystallinity of all N-TiO_2_ samples is similar to that of TiO_2_ after the annealing treatment with urea. This unexpected phenomenon suggests that N-doping hampers the further crystallization of N-TiO_2_ which may enlarge its specific surface area [3,36]. In addition, new diffraction patterns appear close to the (101) crystal plane of N-TiO_2_ (1:2), suggesting that a low amount of urea is enough for the N-doping.

The BET surface areas of all samples were measured to further analyze the effect of the annealing treatment on TiO_2_. From Table 1, it is clear that the BET surface area of hydrothermally prepared TiO_2_ (134.67 m^2^/g) is the largest whereas that of A-TiO_2_ is the smallest (40.21 m^2^/g). Compared with TiO_2_, the specific surface areas of N-TiO_2_ samples decrease and a decreasing tendency is observed as the amount of urea increases. Clearly, the annealing treatment of TiO_2_ with urea inhibits its crystallization which is in agreement with the results of XRD (Figure 3). For the moment, the reason for this phenomenon has been reported and it may result from the N-doping induced by the annealing treatment [3]. This phenomenon is also observed in the P25 samples obtained after the annealing treatment with urea (Appendix A). As a result, the annealing treatment with urea offers a facile approach for the preparation of N-doped TiO_2_ with more active sites reserved.

To further investigate the influences of the annealing treatment on the crystal structure of TiO_2_, EPR spectra of all samples were recorded and are shown in Figure 4. As shown in Figure 4, the characteristic peak with a g value of 2.002 corresponds to Vo [20,37]. Compared with TiO_2_, A-TiO_2_ contains a certain amount of Vo which may result from the both the crystallization of TiO_2_ and the pyrolysis of the organic solvents [25,26]. With the presence of urea, more Vo is introduced into the N-doped TiO_2_ and its amount increases first and then decreases as the amount of urea increases. This result may be due to the N-doping process which not only induces the formation of Vo but can also occupy the Vo. The formation of Vo in N-doped TiO_2_ may contribute to the separation of photogenerated charge carriers [38,39]. Thus, the annealing treatment with urea not only induces the N-doping to extend the photoresponse range of TiO_2_ but also creates Vo in N-doped TiO_2_ which contributes to the separation of photogenerated charge carriers, which could both favor the enhancement of the photocatalytic performance of TiO_2_.

### 3.2. Photocatalytic Performance

The photocatalytic performance of all TiO_2_ photocatalysts was evaluated by the photocatalytic degradation of MO under visible light irradiation. From Figure 5, all TiO_2_ photocatalysts are capable of adsorbing a certain amount of MO and N-TiO_2_ (1:2) shows (1:1) the highest adsorption capacity for MO (12.9%). Compared with TiO_2_, the specific surface area of N-TiO_2_ (1:2) is smaller and its improved adsorption capacity of MO may be due to the formation of functional groups, which is in agreement with the XRD analysis (Figure 3). During visible light irradiation, both TiO_2_ and A-TiO_2_ are not able to degrade MO because they do not absorb the visible light (Figure 2a). With N-TiO_2_ as the photocatalyst, the photocatalytic degradation of MO is feasible and among the samples N-TiO_2_ (2:1) presents the best photocatalytic performance. The photocatalytic performance of N-TiO_2_ proves that the annealing treatment with urea extends the photoresponse range of TiO_2_ to the visible light range [18,40]. The photocatalytic performance of N-TiO_2_ is comparable to the reported results (Appendix A) which may be ascribed to the formation of Vo and the increased specific surface area [33,39,41,42]. Based on the photocatalytic performance, the optimal weight ratio of TiO_2_ to urea for the preparation of N-TiO_2_ via the annealing treatment is found to be 2:1.

## 4. Conclusions

In summary, a facile strategy was developed for the preparation of N-TiO_2_ via the annealing treatment with urea. On the one hand, the photoresponse of TiO_2_ is extended by N-doping via the annealing treatment with urea as the N source. On the other hand, Vo is introduced into N-TiO_2_ which may contribute to the separation of photogenerated charge carriers. In addition, the specific surface area of N-TiO_2_ is enlarged with the presence of urea during the annealing treatment by inhibiting the crystallization of TiO_2_. Thus, more active sites could be reserved for photocatalytic reactions. All the above favorable aspects induced by the annealing treatment with urea contribute to the excellent photocatalytic performance of N-TiO_2_. This facile strategy is also suitable for other TiO_2_ photocatalysts such as P25 and, thus, our work offers a universal approach for the preparation of N-doped TiO_2_ via the annealing treatment with urea. The annealing treatment with other additives for different elements doping, not merely N doping, may be also possible and further work is underway.

## Figures and Tables

**Figure 1 nanomaterials-14-00818-f001:**
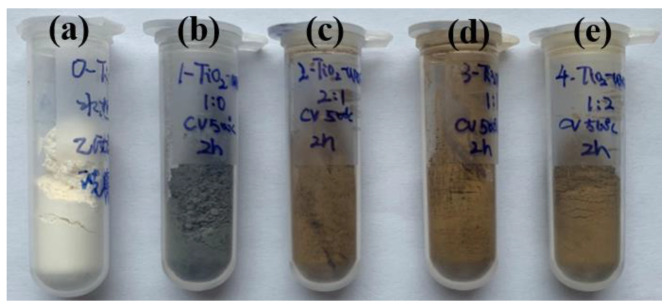
Digital photographs of (**a**) TiO_2_, (**b**) A-TiO_2_, (**c**) N-TiO_2_ (2:1), (**d**) N-TiO_2_ (1:1), and (**e**) N-TiO_2_ (1:2).

**Figure 2 nanomaterials-14-00818-f002:**
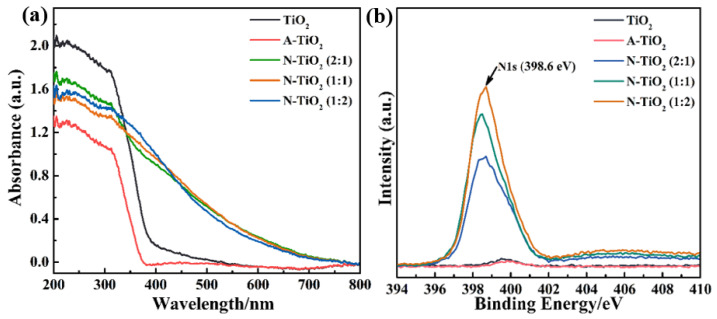
(**a**) DRS spectra and (**b**) N1s XPS fine spectra of TiO_2_, A-TiO_2_, N-TiO_2_ (2:1), N-TiO_2_ (1:1), and N-TiO_2_ (1:2).

**Figure 3 nanomaterials-14-00818-f003:**
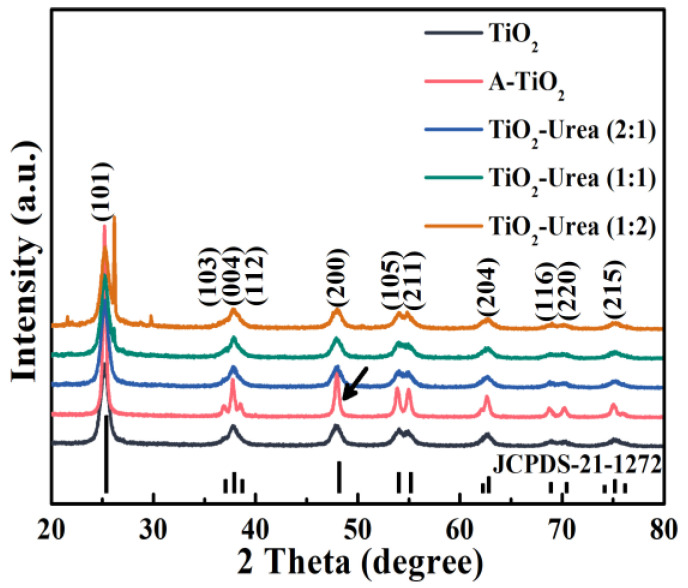
XRD patterns of TiO_2_, A-TiO_2_, N-TiO_2_ (2:1), N-TiO_2_ (1:1), and N-TiO_2_ (1:2). The arrow shows the (200) diffraction peak of A-TiO_2_ [25].

**Figure 4 nanomaterials-14-00818-f004:**
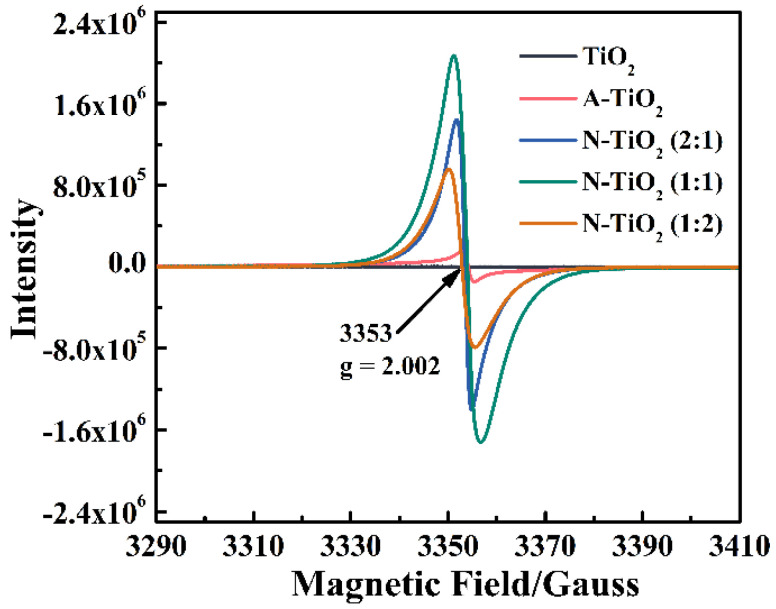
EPR spectra of TiO_2_, ATiO_2_, N-TiO_2_ (2:1), N-TiO_2_ (1:1), and N-TiO_2_ (1:2).

**Figure 5 nanomaterials-14-00818-f005:**
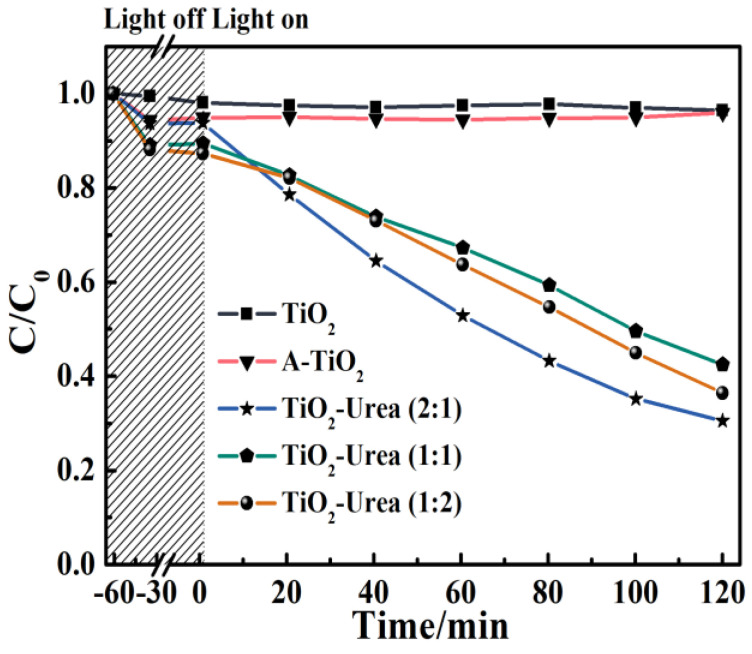
Photocatalytic degradation curves of MO by TiO_2_, A-TiO_2_, N-TiO_2_ (2:1), N-TiO_2_ (1:1), and N-TiO_2_ (1:2) under visible light irradiation.

**Table 1 nanomaterials-14-00818-t001:** BET surface area of TiO_2_, A-TiO_2_, N-TiO_2_ (2:1), N-TiO_2_ (1:1), and N-TiO_2_ (1:2).

Sample	BET (m^2^/g)
TiO_2_	134.67
A-TiO_2_	40.21
N-TiO_2_ (2:1)	107.59
N-TiO_2_ (1:1)	73.36
N-TiO_2_ (1:1)	51.87

## Data Availability

Data will be made available on request.

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
