# Peer review of "A Facile Strategy for the Preparation of N-Doped TiO2 with Oxygen Vacancy via the Annealing Treatment with Urea"

_nanomaterials, 2024, doi:10.3390/nano14100818_

Round 1

Reviewer 1 Report

Comments and Suggestions for Authors

Titanium dioxide (TiO2) is widely used in various applications. Both oxygen-vacancies and doping are commonly used modifications to enhance the photocatalytic performance of TiO2. This work used urea annealing treatment to prepare N-doped TiO2 with oxygen vacancies. After careful considerations, major changes need to be made before the acceptance for Nanomaterials.

1. Authors need to state the novelty of this method in Introduction. What is the advantage of this method compared to previous methods?

2. Section 2.2 Annealing treatment requires more details. Is grinding process involved before annealing? If so, how long is the grinding process?

3. What is the yield of A-P25, N-P25 (2:1), N-P25 (1:1), and N-P25 (1:2) after annealing?

4. Every experiment results lack sufficient discussion.

5. Why is the BET surface area decreased after annealing? Why is the surface area different while changing the amount of urea? More explanations are needed regarding Table 1 and Table S1 results.

6. In Figure 4, why is the amount of oxygen vacancies increased first and then decreased with the increasing of urea amount?

7. Please combine the EPR results to analyze Photocatalytic performance of the samples. Which is more important for photocatalytic performance, nitrogen doping or oxygen vacancy?

8. Please adjust the figures to same size (if possible). Figures in Supporting Information are either too big or too small.

9. Authors need to pay attention to the details. For example, Line 122: letter sizes are different; Line 179: 'as the N source; On the other hand' ; Figures & captions are not centered and aligned.

10. The similarity percentage (35%) is too high according to iThenticate. Please revise carefully.

Comments on the Quality of English Language

Minor modifications should be made in English Language. For example, Line 40: 'could traps'

Reviewer 2 Report

Comments and Suggestions for Authors

In the course of their work, the authors produced a TiO2 catalyst modified with N. In the first step, the catalyst was prepared solvometrically, then the modification was done by annealing.
After reading the article, my questions and comments are as follows:

1.”Although the amount of urea increases, the color of all N-TiO2 samples are similar implying that a low amount of urea is enough for N-doping (Figure 1c-e).” - Has the nitrogen content of the samples been determined?

The surface binding of the model compound develops interestingly:

on the catalyst with the largest specific surface area (TiO2) it is very minimal, while on the samples with a smaller specific surface area (N-TiO2 1:1, 1:2) it is greater - how can this phenomenon be explained?

The surface binding of the model compound develops interestingly:

on the catalyst with the largest specific surface area (TiO2) it is very minimal, while on the samples with a smaller specific surface area (N-TiO2 1:1, 1:2) it is greater - how can this phenomenon be explained?

  Has the reproducibility of the sample preparation been investigated?

Has the stability of the catalysts been investigated? How does the activity change in case of reuse?

Perhaps it would have been better to use a colorless compound as a model compound. Sensitization cannot be ruled out when using a colored compound.

Comments on the Quality of English Language

The language of the article is correct.

Round 2

Reviewer 1 Report

Comments and Suggestions for Authors

The revised draft is improved a lot. It is recommended to be accepted byNanomaterials.

Author Response

Comments:The revised draft is improved a lot. It is recommended to be accepted byNanomaterials.

Author replay:We thank the reviewer 1 for your kind help and appreciate your recognition of our manuscript.

Reviewer 2 Report

Comments and Suggestions for Authors

The authors answered my questions. I accept the corrections. I recommend publishing the article in this form.

Author Response

Comments:The authors answered my questions. I accept the corrections. I recommend publishing the article in this form.

Author replay:We thank the reviewer 2 for your helpful comments and appreciate your acceptance of our manuscript.